# Diagnostic Delays and Economic Burden in Japanese Women with Endometriosis: A Cross-Sectional Analysis

**DOI:** 10.3390/ijerph22111623

**Published:** 2025-10-25

**Authors:** Nobuo Nishimata, Satomi Sato

**Affiliations:** Graduate School of Public Health, St. Luke’s International University, Tokyo 104-0044, Japan

**Keywords:** endometriosis, diagnostic delay, healthcare behavior, women’s health, economic burden, Japan

## Abstract

Background: This study investigates the association between diagnostic delay (DD) and clinical and behavioral variables among Japanese women with endometriosis, and explores an optimal cut-off point distinguishing short and long DD. Methods: a cross-sectional online survey was conducted among 220 Japanese women aged 18–49 diagnosed with endometriosis. Data on healthcare behaviors, economic expenditures, and disease-specific outcomes were analyzed by stratifying participants based on DD length. Multivariate logistic regression models were applied. Results: the mean age at initial symptom onset was 24.3 years, and at diagnosis, 27.7 years. The median DD was 1.5 years, with significant differences between short and long DD groups (*p* < 0.001). Longer DD was significantly associated with greater use of over-the-counter (OTC) pain medication (*p* = 0.008) and a higher proportion of Stage IV endometriosis (*p* = 0.022). Conclusions: diagnostic delays longer than 1.5 years may contribute to disease progression and reliance on self-management, potentially postponing medical consultation. Early intervention strategies, including screenings and public awareness, may promote timely healthcare-seeking behavior. Future studies should prioritize clinical assessments and early diagnosis to reduce the burden of advanced disease.

## 1. Introduction

Hormonal imbalances affect women’s health across life stages, leading to symptoms such as abdominal pain, severe menstrual cramps, and heavy bleeding [1,2]. These conditions may lead to infertility and psychological impacts [3,4,5].

Despite the significance of these issues, symptoms like dysmenorrhea are often overlooked in Japan, contributing to a long-term burden [6]. The Ministry of Economy, Trade and Industry [7] estimates that productivity loss due to these symptoms amounts to 3.4 trillion yen annually. Approximately one-third of dysmenorrhea cases progress to endometriosis, which affects around 10% of reproductive-age women and is strongly associated with infertility [8,9,10,11]. Diagnosis of endometriosis remains challenging, with an average global delay of 6–12 years [12,13], while Japan-specific data remain limited [14].

Diagnostic delay for endometriosis—the time from the onset to confirmed diagnosis—can lead to physical, psychological, and financial impacts throughout woman’s life. These health conditions often lead to irregular absences caused by fatigue, resulting in students missing academic activities, which then disrupts their educational progress and academic achievements. In Japan, menstrual-related symptoms, such as dysmenorrhea, are often overlooked due to cultural misconceptions that “menstrual pain is natural and should be endured” [15,16]. This contributes to delay in seeking consultation with a gynecologist (79.6%) and tolerating pain without taking any over-the-counter (OTC) drugs (67.0%), or preferring to take an OTC painkiller (35.9%) only when they have severe pain [15]. This behavior, in turn, exacerbates the progression of dysmenorrhea to endometriosis. A study from Sweden highlighted how symptoms of dysmenorrhea and their impact on education, which often begins in adolescence, severely continue to hinder educational progress and career choices, directly affecting long-term economic outcomes [17].

A standardized definition to distinguish between “short” and “long” diagnostic delay in endometriosis is still lacking.

Surrey et al. (2020) [18] proposed a three-tier classification: short (≤1 year), intermediate (1–3 years), and long (3–5 years), and found that longer delays were associated with greater clinical burden and higher healthcare costs.

More recently, Brandes et al. (2022) [13] identified a median delay of 5 years when differentiating between Short Diagnostic Delay (Short DD) and Long Diagnostic Delay (Long DD). They found that the most significant predictors of longer delay were younger age at symptom onset (*p* < 0.001) and younger age at first gynecological consultation (*p* = 0.01). These findings suggest that younger women may be particularly vulnerable to delayed diagnosis, which could lead to increased disease burden and economic impact.

Based on these findings, it is critical to address the impacts of diagnostic delay for endometriosis. Delayed therapeutic intervention after symptom onset may negatively affect women’s education and career. This approach has not been previously considered, and it is necessary to investigate whether diagnostic delays contribute to adverse outcomes in education, career development, and economic burden within the Japanese healthcare system.

## 2. Materials and Methods

### 2.1. Study Design

This cross-sectional study aimed to (1) examine the association between diagnostic delay in endometriosis and patient characteristics, and (2) determine the mean and median time from symptom onset to confirmed diagnosis.

An online survey was administered from September to October 2024 among Japanese women aged 18 to 49 years who had been diagnosed with endometriosis. Participants were recruited through Freeasy, an online panel developed by iBridge Co., Ltd. (Osaka, Japan), which is widely used in health-related research to identify individuals diagnosed with endometriosis, being diagnosed in approximately 10% of reproductive women. Considering that 72% of Japanese women were eligible for individual income (according to the employment-to-population ratio of women aged between 15 and 64 [12]), we anticipated conducting a screening survey with 5000 participants to identify 360 eligible respondents, accounting for an expected 20% missing response rate [13]. All participants were required to read a study information sheet and provide informed consent before participating. Only individuals who provided consent were allowed to complete the self-administered and anonymous questionnaire.

Informed by previous findings on diagnostic delay and disease [17], we investigated trends in education, individual income, and expenditures among patients with early symptoms of endometriosis, including dysmenorrhea, pelvic pain, fatigue, and nausea. Analyses included descriptive statistics on endometriosis outcomes and economic status. The endometriosis-specific outcomes included healthcare behavior and experience of endometriosis patients. The collected data were subsequently analyzed to explore factors associated with diagnostic delay and economic and psychosocial impacts.

### 2.2. Participants and Sampling Strategy

Eligibility criteria were as follows:(1)Women aged 18–49 years who provided informed consent at the time of survey participation;(2)Self-reported diagnosis of endometriosis confirmed by laparoscopy, laparotomy, or clinical symptomatic condition;(3)Having independent personal income;

Participants meeting these criteria were recruited from the online survey panel. This study aimed to recruit 220 participants diagnosed with endometriosis. This sample size was calculated based on a previous study in Germany, which included 157 participants for validating diagnostic delay cut-offs [13]. To account for differences in population size between reproductive-age women (15–49 years) in Germany (15.65 million) and Japan (22.94 million), the sample was adjusted using a ratio of 1:1.39. Considering an expected 20% rate of ineligible or incomplete responses, we set the recruitment target at 275 participants to ensure sufficient statistical power.

### 2.3. Ethics Statement

Ethical approval for this study was obtained from the Institutional Review Board of St. Luke’s International University (approval number: 24-R093). All participants provided informed consent online prior to participation, and the survey was conducted anonymously and followed the Declaration of Helsinki, as well as the Ethical Guidelines for Life Science and Medical Research Involving Human Subjects issued by the Government of Japan.

### 2.4. Survey Instrument and Development

The questionnaire in the present study was adapted from that used in the EndoCost study [19], including the version applied by Brandes et al. (2022) [13]. The study evaluated the economic burden of endometriosis across multiple centers in ten countries [19]. The aim of the study was to gather information on a wide range of disease-specific parameters, healthcare costs and health-related quality of life of endometriosis patients from a societal perspective. The original 30-page questionnaire included items such as age at symptom onset, first consultation, age at diagnosis, income, healthcare costs, and other disease-related factors [20].

### 2.5. Questionnaire Content

A 21-item questionnaire was developed with reference to the EndoCost study [19,20], covering sociodemographic and disease-specific parameters tailored to the Japanese context.

In Section 1, participants provided demographic information, such as age, educational attainment, occupation, personal annual income, area of residence, marital status with or without child. Additionally, participants were asked to report the following questions regarding expenditures related to endometriosis:Medical expenditure

“What is your average monthly medical expenses covered by public insurance over the past six months for the treatment of endometriosis at clinics or hospitals?”
2.Transfer fee

“What is the average monthly amount spent on transportation costs (parking fees, gasoline expenses, etc.) for visits to medical institutions (clinics or hospitals)?”
3.Self-care fee

“How much do you spend on self-care for endometriosis?”

In Section 2, participants were asked endometriosis-specific questions, such as awareness of symptoms with/without OTC drugs, age of initial symptom onset, first consultation with gynecologist and confirmed diagnosis of endometriosis, disease stage (if applicable), consultation person, pain intensity using a face scale, and experience of stigma regarding endometriosis.

### 2.6. Variable Classification and Analytical Plan

Participants were categorized into Short DD (<1.5 years) and Long DD (≥1.5 years) groups based on the median diagnostic delay, and associations with healthcare behavior-related variables were subsequently examined.

### 2.7. Statistical Analysis

Descriptive statistical analysis was conducted for all variables [8,13,21]. Between-group comparisons were performed using *t*-tests for continuous variables and chi-square tests for categorical variables. Cases with missing responses were excluded from the analysis [13]. Multiple linear regression analyses were performed to assess associations between diagnostic delay, social stigma, and economic burden. All analyses were conducted using STATA software (version 18.0; StataCorp LLC, College Station, TX, USA).

## 3. Results

### 3.1. Study Population

A total of 5000 women were initially screened to confirm a diagnosis of endometriosis. Among these, 371 women met the diagnostic criteria for endometriosis. Subsequently, eligibility criteria were applied, which required participants with individual income, leading to the exclusion of 48 individuals who did not meet this criterion. This process resulted in a preliminary cohort of 323 eligible participants. Of these, a total of 287 women with endometriosis completed the study questionnaire, corresponding to a response rate of 88.9%. Fourteen of these were excluded due to missing or unclear diagnostic delay data, and 53 participants with asymptomatic were also excluded; finally, a total of 220 endometriosis patients who reported experiencing initial symptoms before being diagnosed with endometriosis participated (Figure 1).

For the target population with symptoms, the mean age was 36.1 years old, with the majority falling in the age group of 30–40 years old (38.2%). Regarding marital status, 53.2% were married, 70.1% of whom had at least one child; 46.8% were unmarried, 8.7% of whom had at least one child, indicating the single-parent rate. In terms of annual income, the average annual income was 3.7 million JPY among this population. The Japanese average personal annual income is 4.6 million JPY, which means 0.9 million JPY less among study population. Medical costs were calculated based on categorical variables derived from respondent-selected choices to account for variations; the average monthly medical expenditure calculated from the mean of the responses from 1 to 7 (mean 2.58) was JPY 5160.

### 3.2. Endometriosis Specific Outcomes

Regarding endometriosis symptoms, it was essential to identify the timing of initial symptom onset and the age at diagnosis to characterize diagnostic delay. Differences between the diagnostic delay groups represent novel findings in the Japanese context.

Table 1 presents the results of the primary objective which showed the distribution of endometriosis specific outcomes and healthcare behavior. The difference in mean between the age of initial symptom onset confirmed diagnostic age was 3.6 years, which was a similar to China. Following consultation with a gynecologist, patients experienced a 1.3-year diagnostic delay for endometriosis from initial onset. However, there was a 2.1-year interval between the manifestation of initial symptoms and the consultation with a gynecologist; this difference is indicative of Japanese healthcare behavior. In contrast to a previous study, this population predominantly comprised OTC-drug users. In terms of staging according to the ASRM score, the presence of stage I was the highest at 53.2% and stage IV was the second highest at 36.2%, indicating that staging was bilaterally segregated, although the number of participants was limited to 47 people (21.4%, compared with total population). The social stigma most commonly experienced by endometriosis patients, as indicated by a survey, was the perception that ‘Pain is common’ (25.9%).

### 3.3. Economic Situation of Patients with Endometriosis

Figure 2 presents the regression analysis of predictive margins regarding the economic situation, which was also one of primary objective for endometriosis patients in this study. This figure included the following variables: age, age of symptom onset, and economic expenditures including medical expense, transfer fee, and self-care fee related to endometriosis. According to a previous Swedish research, early symptom onset affected long-term educational and economic outcomes. Therefore, each variable included in the regression analysis model was identified based on the following demographic and economic variables: age, age of initial symptom onset, education, occupation, medical expense, transfer fee, self-care fee, OTC-drug use, and having a child.

Each 95% confidence interval (CI) is shown.

(A)A trend toward higher annual income with older age at symptom onset, although not statistically significant (*p* = 0.474).(B)Significantly higher income among those with graduate-level education compared to high school graduates (*p* = 0.001).(C)Younger participants tended to spend more on self-care fees, with a statistically significant trend (*p* = 0.044).(D)A potential increase in self-care fee with higher annual income (*p* = 0.395), though this association was not significant.

### 3.4. Exploration of Diagnostic Delay Cut-Off Score

The second objective was to identify the mean and median diagnostic delay. The analysis of the initial symptom age and diagnostic age revealed a mean and median diagnostic delay (difference between diagnostic age and initial symptom age: mean = 3.6 years, median = 1.5 years, IQR = [min: 0, max: 24.3], Table 2). Although we attempted to identify the diagnostic-delay group as a mean of two-category approach based on prior studies, the diagnostic delay groupings were inappropriate for comparative analysis across study arms. Therefore, the median value of 1.5 years was used to classify participants into the Short DD and Long DD groups as outlined by Brandes et al. (2022) [13]. Participants with a diagnostic delay of ≤1.5 years were categorized as the Short DD group, while those with a delay of >1.5 years were categorized as the Long DD group.

### 3.5. Association of Diagnostic Delay and Variables of Characteristics

As part of the analysis plan, comparisons between diagnostic delay groups (Short DD and Long DD) in terms of health behavior, education, occupation, income, and economic expenditure were conducted as shown in Table 3. The *p*-values presented in this table are the results of univariate analysis, which assessed the statistical differences between the Short DD and Long DD groups for each variable. The results of the initial symptom age and first consultation age with gynecologist were significantly earlier age in the Long DD group (*p* < 0.001, *p* = 0.003). The difference between initial symptom onset age and diagnosis age was statistically significant among the Short DD and Long DD groups (*p* < 0.001). Individuals in the Long DD group were more likely to use OTC drugs (*p* = 0.011). The Long DD group also had a significantly higher marriage rate (*p* = 0.037). Expenditure of self-care fee was significantly higher in the Short DD group (*p* < 0.001).

Progression from symptom onset to receiving a definitive diagnosis by a specialist is depicted for both the Short DD and Long DD groups in Figure 3. The time scale of difference between age at symptom onset and age at consultation with a gynecologist was 2.7 years for Long DD and 1.3 years for Short DD (*p* = 0.02).

Endometriosis-related characteristics for each diagnostic delay group indicated that the distribution of endometriosis stages differs significantly between the Short DD and Long DD groups (*p* = 0.049). Stage I was more commonly observed in the Short DD group, whereas Stage IV was predominant in the Long DD group. On the other hand, the ASRM scores showed a higher trend in the Long DD group than the Short DD group (37.6 vs. 19.9) although the difference did not reach statistical significance (*p* = 0.072). Consultation with medical staff showed a higher trend in the Long DD group, but it was not significantly different (*p* = 0.065). The face scale, which measures distress on a scale from 1 to 10, was significantly higher in the Long DD group than the Short DD group (4.16 vs. 3.48, *p* < 0.001).

Logistic regression was conducted to investigate factors associated with diagnostic delay (Short DD vs. Long DD). Stepwise regression was performed to select variables related to association between diagnostic delay and characteristics, using a significance threshold of *p* < 0.05 to determine the model. The results of the logistic regression analysis using the selected variables are presented in Table 4.

The results revealed that the odds of experiencing a long diagnostic delay decreased by 5% with every one-year increase in age at first consultation with a gynecologist, which was statistically significant (OR = 0.95, 95% CI = 0.92–0.98, *p* = 0.002). The results also revealed the odds of OTC drug using was significantly higher in the Long DD group (*p* = 0.008). Participants with a Junior College education, had significantly higher odds of experiencing a long diagnostic delay compared to high school/vocational school (OR = 3.60, 95% CI = 1.36–9.53, *p* = 0.010). In contrast, participants with medium expense group (5000–10,000 JPY/month) were likely to experience a short diagnostic delay compared with lower expense group, less than 3000 JPY/month and 3000–5000 JPY/month (OR = 0.27, 95% CI = 0.11–0.66, *p* = 0.003). No other factors, including occupation, annual income, self-care fees, and transfer fees, were significantly associated with diagnostic delay.

Table 5 shows the results of logistic regression analysis examining the factors from each variable associated with Endometriosis-specific outcomes (Short DD vs. Long DD). Stepwise regression was performed to select variables related to diagnostic delay and endometriosis specific outcomes, using a significance threshold of *p* < 0.05 to determine the model. The results of the logistic regression analysis using the selected variables revealed that those in Stage 4 of endometriosis had significantly higher odds of long diagnostic delay than those in Stage 1 (OR: 4.71, 95% CI = 1.25–17.79, *p* = 0.022).

Figure 4 shows the results of panels A–D illustrate predictive margins with 95% confidence intervals to compare economic indicators between Short and Long diagnostic delay (DD) groups, adjusted for relevant covariates.(A)Initial symptom onset age and annual income: While not statistically significant (*p* = 0.455), those with earlier symptom onset in the Long DD group tended to have lower annual income than their Short DD counterparts.(B)Education and annual income: Across all education levels, the Long DD group showed lower income predictions, with the largest disparity observed among graduate school attendees. However, none of the differences reached statistical significance.(C)Study entry age and self-care expenses: In both groups, younger participants showed higher self-care costs, particularly in the Short DD group. This trend declined with age, although differences between groups were not significant (*p* = 0.863).(D)Annual income and self-care expenses: In the Short DD group, self-care expenses increased with income level, whereas the Long DD group showed a relatively flat trend. This contrast approached statistical significance (*p* = 0.063).

Findings from Figure 4 showed that there was no significant difference in economic situation regarding early symptom age, educational attainment, selfcare fee, and annual income. These plots were included to visualize potential trends and uncertainty ranges even in the absence of statistical significance, offering supplementary insights into the economic implications of diagnostic delay.

## 4. Discussion

### 4.1. Key Determinants of Diagnostic Delay

Our study identified several factors significantly associated with prolonged diagnostic delay (Long DD) among patients with endometriosis. Specifically, patients experiencing a delay of more than 1.5 years were more likely to use OTC drugs (OR: 2.36, *p* = 0.008), have a Junior College education (OR: 3.60, *p* = 0.010), and present with Stage IV disease (OR: 4.71, *p* = 0.022). These findings suggest that self-medication may obscure symptom severity, contributing to later diagnosis. Furthermore, healthcare-seeking behaviors differed significantly between Short DD and Long DD groups (*p* = 0.02), potentially reflecting cultural norms that normalize menstrual pain and discourage early consultation by social stigma. Notably, older age at first consultation and higher healthcare expenditures were associated with shorter diagnostic delays—possibly indicating more proactive engagement with the medical system among these individuals.

### 4.2. Cultural and Structural Influences in Japan

Unlike prior studies from Western countries, our study reveals how Japan’s unique sociocultural and healthcare context shapes diagnostic trajectories. Cultural expectations of endurance and modesty surrounding menstruation may lead women to normalize pelvic pain, resulting in delayed gynecological consultations and continued reliance on OTC drugs. These behavioral tendencies are compounded by structural factors such as the absence of a gatekeeping primary care system under Japan’s universal health coverage (UHC), which often leads to fragmented care and inefficient referrals to specialists. These findings highlight the interplay between individual behavior and systemic barriers in diagnostic delays.

### 4.3. Predictive Margins and Socioeconomic Patterns

Although statistical significance was not achieved for associations between early symptom onset and socioeconomic outcomes, the predictive margin plots (Figure 2 and Figure 4) reveal subtle trends. Participants in the Long DD group—particularly those with early symptom onset—tended to report lower annual incomes, echoing earlier findings from Swedish studies [17,22]. Moreover, younger participants consistently reported higher self-care expenditures (*p* = 0.044), suggesting that the economic burden of managing endometriosis disproportionately affects younger age groups. While these trends were not conclusive (*p* values > 0.05 in several models), the visualizations facilitate nuanced interpretation and hypothesis generation, especially in conditions like endometriosis, where social and economic factors intricately interact.

### 4.4. Methodological Justification for Cut-Off Definitions

The diagnostic delay cut-off of 1.5 years was defined using the median to ensure balanced group sizes (*n* = 114 vs. *n* = 106). Although the average delay was longer (3.6 years, SD = 5.2), adopting the mean would have resulted in significant imbalance (Short DD: *n* = 152; Long DD: *n* = 68). This methodological choice aligns with recent recommendations by Brandes et al. (2022) [13] and enhances the validity of between-group comparisons.

### 4.5. Implications and Future Directions

Our study is among the first to explore diagnostic delays in endometriosis within the context of Japan’s UHC framework and sociocultural environment. Our findings offer an important contribution by illuminating the invisible, non-clinical burden borne by patients—especially younger women—who self-medicate while navigating a fragmented care pathway. Despite limitations such as the self-reported nature of the data and lack of longitudinal follow-up, these results provide a valuable reference for international comparisons and policymaking aimed at reducing diagnostic delays and health inequities.

Coordinated strategies—including public awareness campaigns, physician training, and referral system reforms—are urgently needed. These efforts can shorten diagnostic intervals, reduce disease progression, and lessen the socioeconomic impact on patients and families.

## 5. Limitations

Our study has several limitations that should be acknowledged. First, as a cross-sectional study, it cannot establish causal relationships between health behavior and endometriosis. The observed associations merely reflect associations at a single time point, and the temporal relationship remains unclear. Second, the results are self-reported, which introduces potential recall bias. Participants may misremember or misestimate key information, such as healthcare expenditures and potentially affecting the accuracy of the findings. As the survey was conducted via an online, there may be a selection bias favoring individuals with internet access and greater health awareness. This potential bias also should be considered when interpreting the generalizability of the results. Third, the diagnosis of endometriosis itself is also self-reported, lacking confirmation through medical records or clinical evaluation. This might lead to misclassification or overrepresentation of the condition. Furthermore, our study did not incorporate clinical data such as disease staging, treatment history, or consultation records, which limits the clinical precision of the findings. Long-term outcomes were also not assessed. Fourth, potential confounding factors remain unaddressed, because detailed information on variables such as socioeconomic status, comorbidities, and access to healthcare was not collected. These unmeasured confounders could potentially bias the results. Fifth, our study was conducted with a relatively small and limited sample size of 220 participants, and the distribution of participants across prefectures was uneven. Sixth, our study was limited to individuals with independent income, which could introduce bias by excluding those dependent on shared household resources or external financial support. This aspect was not thoroughly analyzed and presents an opportunity for further investigation. To enhance clinical validity and generate more robust evidence, future studies should address these limitations by linking survey data with medical records, adopting longitudinal designs, recruiting larger and more representative samples, and examining how clinical severity affects healthcare expenditures and, consequently, household financial burden, with implications for health policy.

## 6. Conclusions

Our study highlights the multifaceted nature of diagnostic delay in endometriosis, revealing its associations with patient self-management behaviors, educational background, and disease severity. By using a median-based cut-off of 1.5 years to define diagnostic delay, we identified that prolonged delays were particularly associated with reliance on over-the-counter (OTC) medication and younger age at symptom onset—suggesting that pain normalization and sociocultural stigma may deter timely gynecological consultation in Japan.

Our findings underscore the need for early intervention strategies that go beyond clinical measures, including menstrual health education, public awareness campaigns, and structural reforms to improve access to gynecologic care. Importantly, this study provides new context-specific evidence from Japan’s UHC setting, offering a valuable reference for international comparative research on diagnostic inequality in gynecological conditions.

By illuminating hidden patterns in health-seeking behavior and diagnostic trajectories, this research contributes to making the invisible burden of endometriosis more visible—not only for healthcare professionals and policymakers, but also for patients and their families navigating the condition.

## Figures and Tables

**Figure 1 ijerph-22-01623-f001:**
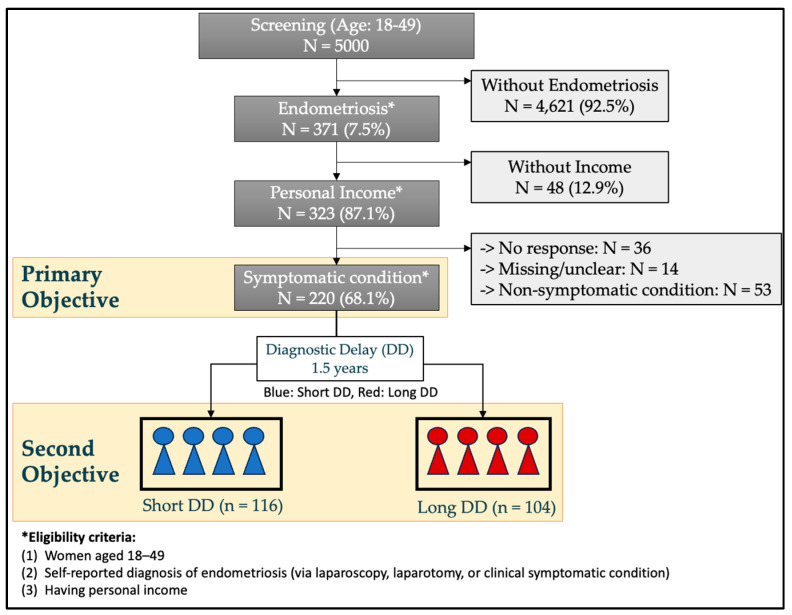
Patient selection process and subgroup allocation based on diagnostic delay.

**Figure 2 ijerph-22-01623-f002:**
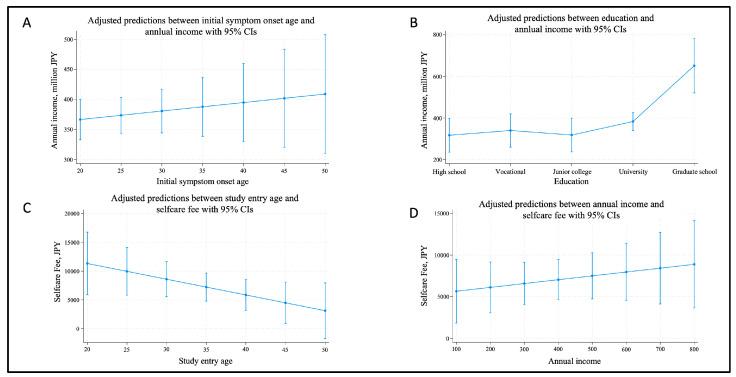
Predictive margins of relationship regarding economic situation for each variable.

**Figure 3 ijerph-22-01623-f003:**
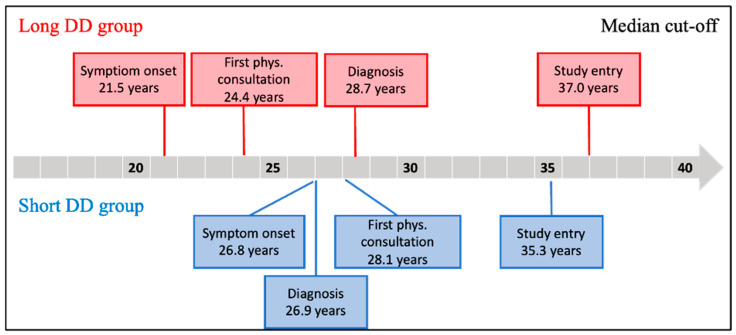
Comparison of the phases for groups with Long DD (red) and Short DD (blue). Each categorical variable was classified by median cut-off, 1.5 years.

**Figure 4 ijerph-22-01623-f004:**
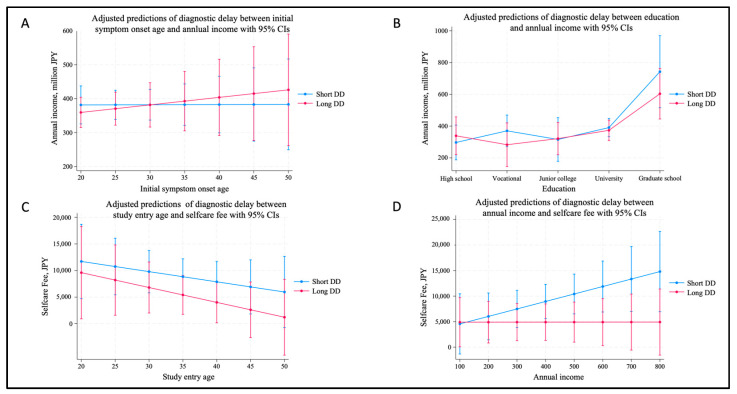
Predictive margins of relationship regarding economic situation on each variable among diagnostic delay groups.

**Table 1 ijerph-22-01623-t001:** Patient characteristics of the study participants with symptoms.

Endometriosis with Symptoms, *N* = 220			
**Study entry age**	** *N (%)* **	** *Mean (SD)* **	** *Range* **
	220 (100)	36.1 (8.1)	18.3–49.7
**Symptom Onset to Diagnosis Timeline**			
Initial symptom onset age	220 (100)	24.3 (8.3)	5.0–47.0
First physician consultation age	220 (100)	26.4 (9.1)	5.1–49.0
Confirmed diagnosis age	220 (100)	27.7 (7.7)	8.6–48.9
Difference between symptom onset age and diagnosis age, mean (SD)	220 (100)	3.6 (5.2)	0–24.3
**OTC drug use**			
With OTC drug	153 (69.5)	1.30 (0.5)	1–2
Without OTC drug	67 (30.5)		
**Stage (*N* = 47)**			
I	25 (53.2)	2.3 (1.4)	1–4
II	2 (4.3)		
III	3 (6.4)		
IV	17 (36.2)		
**ASRM score**	47 (100)	27.1 (33.1)	1–99
**Consultation person, multi-selective option (*N* =220)**			
Medical staff	121 (35.9)	1 (0)	1–7
Family	75 (22.3)		
Partner	69 (20.5)		
Friend	25 (7.4)		
Colleague	15 (4.5)		
Teacher	4 (1.2)		
No person	28 (8.3)		
Face scale			
1 (no pain)–10 (severe pain)	220(100)	3.8 (1.5)	1–6
**Social stigma, multi-selective option (*N* =220)**			
Pain is common	57 (25.9)	1 (0)	1–8
Experience of ridiculed	52 (23.6)		
Not disclosed	25 (11.4)		
Endured sex with partner	55 (25.0)		
Suspected pseudo-sickness	34 (15.5)		
Not having motivation	45 (20.5)		
Pretend normal	50 (22.7)		
Label complainer	9 (4.1)		

Note: Minimum of difference between diagnostic age and initial symptom onset age was adjusted to zero when the net result was negative.

**Table 2 ijerph-22-01623-t002:** Cut-off score between Short DD and Long DD.

Items	Obs	Mean	Std. Dev.	Min	50%	Max
Initial symptom onset age	220	24.3	8.31	5.0	24.0	47.0
Diagnosis age		27.7	7.70	8.6	26.9	48.9
Difference btw diag and symp age		3.6	5.20	0.0	1.5	24.3

Note: Minimum of difference between diagnostic age and initial symptom onset age was adjusted to zero when the net result was negative.

**Table 3 ijerph-22-01623-t003:** Patient characteristics between Short DD and Long DD groups.

*N*	Short DD	Long DD	*p* Value
Number of participants, *n* (%)	116 (52.7)	104 (47.3)	
Study entry age, mean (SD)	35.3 (8.4)	37.0 (7.7)	0.118
Initial symptom onset age, mean (SD)	26.8 (7.7)	21.5 (8.1)	**<0.001 *****
First physician consultation age, mean (SD)	28.1 (8.6)	24.4 (9.3)	**0.003 ****
Confirmed diagnosis age, mean (SD)	26.9 (7.9)	28.7 (7.3)	**0.089 †**
Difference between symptom onset age and diagnosis age, mean (SD)	0.1 (0.2)	7.2 (0.6)	**<0.001 *****
Difference between symptom onset age and first physician consultation age, mean (SD)	1.3 (0.4)	2.9 (0.6)	**0.02 ***
With OTC-drug, *n* (%)	72 (62.1)	81 (77.9)	**0.011 ***
Without OTC-drug, *n* (%)	44 (37.9)	23 (22.1)	
**Age group, *n* (%)**			
18–19 years old	0 (0)	1 (1.0)	**0.052 †**
20–29 years old	39 (33.6)	19 (18.3)	
30–39 years old	39 (33.6)	45 (43.3)	
40–49 years old	38 (32.8)	39 (37.5)	
**Education, *n* (%)**			
High school	17 (14.7)	15 (14.4)	**0.091 †**
Vocational school	21 (18.1)	11 (10.6)	
Junior college	11 (9.5)	20 (19.2)	
University	63 (54.3)	50 (48.1)	
Graduate school	4 (3.5)	8 (7.7)	
**Employment status, *n* (%)**			
Full-time worker	83 (71.6)	68 (65.4)	0.417
Temporary employee	28 (24.1)	33 (31.7)	
Other/Freelancer	5 (4.3)	3 (2.9)	
**Resident area, *n* (%)**			
Rural area	41 (35.3)	35 (33.7)	0.792
Urban area	75 (64.7)	69 (66.4)	
**Marriage, *n* (%)**			
Unmarried	62 (53.5)	41 (39.4)	**0.037 ***
Married	54 (46.6)	63 (60.6)	
**Having child/not, *n* (%)**			
No	71 (61.2)	58 (55.8)	0.414
Yes	45 (38.8)	46 (44.2)	
**Monthly income JPY, *n* (%)**			
less than 150,000	17 (14.7)	29 (27.9)	**0.063 †**
150,000–300,000	46 (40.0)	30 (28.9)	
300,000–450,000	28 (24.1)	28 (27.0)	
450,000–600,000	18 (15.5)	9 (8.7)	
more than 600,000	7 (6.0)	8 (7.7)	
**House annual income level | 10,000 JPY, *n* (%)**			
Low house income (<600)	44 (38.0)	48 (46.2)	0.240
Middle house income (>600, <1200)	60 (51.7)	42 (40.4)	
High house income (>1200)	12 (10.3)	14 (13.5)	
**Monthly medical expenditure JPY, *n* (%)**			
less than 3000	37 (32.0)	39 (37.5)	0.120
3000–5000	18 (15.5)	27 (26.0)	
5000–10,000	24 (20.7)	10 (9.6)	
10,000–15,000	21 (18.1)	14 (13.5)	
15,000–30,000	13 (11.2)	10 (9.6)	
30,000–50,000	1 (0.9)	3 (2.9)	
more than 50,000	2 (1.7)	1 (1.0)	
**Monthly transfer fee (JPY), *n* (%)**			
less than 1000	58 (50.0)	64 (61.5)	0.219
1000–10,000	45 (38.8)	32 (30.8)	
more than 10,000	13 (11.2)	8 (7.7)	
**Monthly self-care fee (JPY), *n* (%)**			
less than 1000	38 (32.8)	61 (58.7)	**<0.001 *****
1000–10,000	64 (55.2)	33 (31.7)	
more than 10,000	14 (12.1)	10 (9.6)	
**Expense ratio group, *n* (%)**			
less than 5%	65 (56.0)	63 (60.6)	0.776
5–10%	29 (25.0)	21 (20.2)	
10–15%	9 (7.8)	10 (9.6)	
more than 15%	13 (11.2)	10 (9.6)	
**Stage, *n* (%)**			
I	18 (64.3)	7 (36.8)	**0.049 ***
II	1 (3.6)	1 (5.3)	
III	3 (10.7)	0 (0)	
IV	6 (21.4)	11 (57.9)	
**ASRM score, mean (SD)**	19.9 (5.9)	37.6 (7.8)	**0.072 †**
**Consultation person, *n* (%)**			
Medical staff	57 (49.1)	64 (61.5)	**0.065 †**
Family	39 (33.6)	36 (34.6)	0.877
Partner	36 (31.0)	33 (31.7)	0.912
Friend	14 (12.1)	11 (10.6)	0.728
Colleague	6 (5.2)	9 (8.7)	0.306
Teacher	2 (1.7)	2 (1.9)	0.912
No person	14 (12.1)	14 (13.5)	0.757
**Face scale | 1–10, mean (SD)**	3.48 (1.5)	4.16 (1.4)	**<0.001 *****
**Social stigma, *n* (%)**			
Pain is common	32 (27.6)	25 (24.0)	0.549
Ridiculed	24 (20.7)	18 (17.3)	0.524
Not disclosed	15 (12.9)	10 (9.6)	0.439
Experience of dyspareunia	34 (29.3)	21 (20.2)	0.119
Feigned	18 (15.5)	16 (15.4)	0.978
Not having motivation	21 (18.1)	24 (23.1)	0.361
Pretend normal	22 (19.0)	28 (27.0)	0.160
Label complainer	4 (3.5)	5 (4.8)	0.611

Note: Area of residence was categorized as urban area (prefectures where 10 or more participants come from and rural area prefectures where 9 or fewer participants come from Table A1). Expense ratio group was calculated by combining the ratio for annual income with the amount of annual expenditure of medical expense, transfer fee, and self-care fee. *p* < 0.1 †, *p* < 0.05 *, *p* < 0.01 **, *p* < 0.001 *** are highlighted in bold.

**Table 4 ijerph-22-01623-t004:** Logistic regression analysis regarding association between diagnostic delay and characteristics.

Diagnostic Delay (0: Short DD; 1: Long DD)	Odds Ratio (OR)	95% CI	*p*-Value
**First physician consultation age**	0.95	0.92–0.98	**0.002 ****
**OTC drug use**			
No OTC drug use	Ref		
OTC drug use	2.36	1.25–4.45	**0.008 ****
**Education**			
High school/Vocational school	Ref		
Junior college	3.60	1.36–9.53	**0.010 ****
University/Graduate school	1.50	0.76–2.97	0.239
**Medical expenditure**			
Lower expense	Ref		
Medium expense	0.27	0.11–0.65	**0.003 ****
Higher expense	0.54	0.27–1.06	**0.073 †**

Note: High school/vocational school was the reference in education category. Lower expense was the reference in medical expenditure category. Lower expense indicates “less than 3000” and “3000–5000” JPY. Medium expense indicates “5000–10,000” JPY. Higher expense indicates more than 10,000 JPY. *p* < 0.1 †, *p* < 0.01 ** are highlighted in bold.

**Table 5 ijerph-22-01623-t005:** Logistic regression analysis regarding endometriosis specific outcomes between Short DD and Long DD.

Diagnostic Delay (0: Short DD, 1: Long DD)	Odds Ratio (OR)	95% CI	*p*-value
**Stage**			
1	Ref		
2	2.57	0.14–47.02	0.524
3	1	-	**-**
4	4.71	1.25–17.79	**0.022 ***

Note: Stage 1 was the reference in Stage category. *p* < 0.05 * are highlighted in bold.

## Data Availability

The datasets generated and analyzed during the current study are not publicly available due to ethical and privacy considerations. Participants were explicitly assured that their responses would remain anonymous and confidential, in accordance with the approved research protocol and institutional ethics guidelines. However, de-identified data may be made available from the corresponding author upon reasonable request, subject to approval by the Institutional Review Board of St. Luke’s International University.

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
