# Peer review of "Diagnostic Delays and Economic Burden in Japanese Women with Endometriosis: A Cross-Sectional Analysis"

_ijerph, 2025, doi:10.3390/ijerph22111623_

Round 1

Reviewer 1 Report

Comments and Suggestions for Authors

This article presents an interesting study focused on the sociologically relevant issue of endometriosis. The research surveyed a Japanese population of reproductive-age women using a 21-item questionnaire. This instrument covered demographic details, self-care costs, awareness of endometriosis symptoms (including pain), the timeline of first gynecological consultation and diagnosis, and the stage of the disease.

The article's main strengths lie in its well-designed study group and methodology, including the statistical analysis provided.

The following points address limitations and suggest areas for improvement:

1. Limited Generalizability: The study focuses solely on the Japanese population, which narrows the scope of the findings. The results cannot be broadly generalized to the wider population.

2. Lack of Clinical Data: The results rely exclusively on survey-derived data and lack clinical associations. Furthermore, the study does not provide information on long-term outcomes.

3. Focus of Investigation: While the investigation highlights the economic and educational burden associated with delayed endometriosis diagnosis, it would be most interesting to see a deeper analysis of [**Suggestion for a more interesting focus, e.g., the specific mechanisms driving these delays or policy recommendations**].

4. Novelty: Although the results are interesting, they are not groundbreaking and do not significantly contribute new findings to the existing body of literature.

5. Figure Presentation: Figures 1 and 3 display statistically insignificant results. There is a lack of explanation for these results in the figure captions, and all abbreviations/shortcuts should be clearly defined.

6. Discussion Section: The discussion lacks emphasis on the novelty and innovation of the study.

7. Language and Style: The current text includes repetitive and overly long phrases. The discussion needs to be more concise and analytical, with a strong emphasis on the main findings and future perspectives of the research.

Author Response

Please see the attachment files as follows:

Reviewer 2 Report

Comments and Suggestions for Authors
  1. General concept comments
  • The introduction is well-documented and contains relevant information.
  • The aim is clearly stated.
  • The inclusion criteria are not clear, and the fact that the survey was conducted online may introduce a selection bias. For example, women with internet access and an interest in health are more likely to participate.
  • The ”study population” should be included in the "Materials and Methods” section, and a diagram illustrating patient selection can be included in the ”Results”.
  • The ”Results” section should not contain scientific references, but rather your own descriptive analysis. All cited references should be included in the ”Discussion” section, as comments on the obtained results. If the journal allows the publication of this format, then do not modify it.
  • The tables and figures are clear and suggestive for understanding the text.
  • The results draw attention to an important public health issue and provide data that can underpin interventions. This study would be more valuable if followed by a longitudinal study to monitor patients' progress over time.
  • The statements in the ”Discussion” chapter are coherent and supported by the cited references.
  • The conclusions are very important, even if the study has several limitations. Implementing early intervention measures can promote timely medical consultation and help reduce the impact of advanced-stage disease.
  • The cited references are mostly relatively recent publications (from the last 10-15 years) and are appropriate for the discussed topic.
  1. Specific comments
  • Lines 39-41 – You should rephrase this sentence for better clarity. E.g. ”These health conditions often lead to irregular absences caused by fatigue, resulting in students missing academic activities, which then disrupts their educational progress and academic achievements.”
  • Lines 43-44 – Replace ”endure” with a different word to avoid repetition or rephrase the sentence, for example: ”This contributes to the delay in seeking consultation with a gynecologist (79.6%) and tolerating pain without taking any over-the-counter drugs or...”
  • Lines 50-62 – The paragraph summarizes several literature findings but has a confusing structure, making it too heavy for an introduction. The sentences are too long, and the academic style is lost.
  • Lines 99-100 – You should rephrase the text for clarity: ”Considering an expected 20% rate of ineligible or incomplete responses, we set the recruitment target at 275 participants to ensure adequate statistical power.”
  • Lines 182-188 – You should replace this paragraph in the ”Discussion” section.
  • Line 370 – Replace ”as” with ”because” in the phrase: ”..potential confounding factors remain unaddressed because detailed information ...”
  • Lines 374-375 – Rephrase for better clarity, for example: ”..this study was limited to individuals with independent income, which could introduce bias by excluding those dependent on shared household resources or external financial support.”
Comments on the Quality of English Language

The text deserves a review of the English language and academic style; please follow the suggested recommendations.

Author Response

(The authors gave the same response as above.)
